# In-Depth Insight into the Ag/CNQDs/g-C_3_N_4_ Photocatalytic Degradation of Typical Antibiotics: Influence Factor, Mechanism and Toxicity Evaluation of Intermediates

**DOI:** 10.3390/molecules28041597

**Published:** 2023-02-07

**Authors:** Chen Li, Tianyi Sun, Guohui Yi, Dashuai Zhang, Yan Zhang, Xiaoxue Lin, Jinrui Liu, Zaifeng Shi, Qiang Lin

**Affiliations:** 1Key Laboratory of Water Pollution Treatment and Resource Reuse of Hainan Province, Hainan Normal University, Haikou 571158, China; 2School of Chemistry and Chemical Engineering, Hainan Normal University, Haikou 571158, China; 3Public Research Laboratory, Hainan Medical University, Haikou 571199, China

**Keywords:** Ag/CNQDs/g-C_3_N_4_, antibiotics, photocatalytic, mechanism, toxicity evaluation

## Abstract

In this paper, the photocatalytic degradation efficiency of typical antibiotics (norfloxacin (NOR), sulfamethoxazole (SMX) and tetracycline hydrochloride (TCH)) by Ag/CNQDs/g-C_3_N_4_ under visible light irradiation was studied. Various strategies were applied to characterize the morphology, structure and photochemical properties of the Ag/CNQDs/g-C_3_N_4_ composites. The superior photocatalytic activity of Ag/CNQDs/g-C_3_N_4_ was attributed to the wide light response range and the enhancement of interfacial charge transfer. At the same time, the effect of the influence factors (pH, Humic acid (HA) and coexisting ions) on the antibiotics degradation were also investigated. Furthermore, the electron spin resonance (ESR) technology, free radical quenching experiments, LC/MS and DFT theoretical calculations were applied to predict and identify the active groups and intermediates during the photocatalytic degradation process. In addition, Ag/CNQDs/g-C_3_N_4_ exhibited the obvious antibacterial effect to Escherichia coli due to the addition of Ag NPs. This study develops a new way for the removal of emerging antibiotic pollution from wastewaters.

## 1. Introduction

With the wide application of antibiotics in the medical, veterinary, animal husbandry and other fields, antibiotics pollution has been widely regarded as a major environmental problem worldwide [1,2,3,4]. To solve these problems, several techniques have been developed, including chemical, biological, and physical approaches [5,6,7,8]. Among them, the semiconductor photocatalysis, as a potential method to solve global energy shortage and alleviate environmental pollution, has received much attention in the field of environmental remediation [9,10,11,12]. However, the traditional photocatalytic materials such as TiO_2_ were usually driven by ultraviolet light and had low utilization rate for visible light, thus greatly limiting its practical application [13,14]. Therefore, the exploration of efficient photocatalysts driven by visible light has attracted more and more attention [15,16,17,18] in recent decades.

As a typical photocatalyst responsive to visible light, graphite phase carbon nitride (g-C_3_N_4_) exhibits great application potential in the field of photocatalysis due to its suitable band gap of 2.7 eV, good chemical stability, easy availability of raw materials and environmental friendliness [19]. Unfortunately, similarly to many photocatalysts with a single component, the practical application of g-C_3_N_4_ was limited by the inherent disadvantages, including agglomeration, high carrier recombination rate and low solar energy utilization rate [20]. Therefore, it is necessary to continuously develop the composites based on g-C_3_N_4_ to promote the separation of photo-generated electron–hole pairs and to further improve the photocatalytic performance. Compared with the metal oxide catalysts with rigid structure, g-C_3_N_4_ with the conjugated polymerization structure had some advantages in structural control. Various methods exist to improve the photocatalytic activity of g-C_3_N_4_, including nano-structure construction [21,22,23,24,25,26], structural defect modification [27,28], surface property modification [2], and construct heterostructures [29,30]. The enhanced photocatalytic activity of modified g-C_3_N_4_ could be mainly attributed to the increase in specific surface area, effective separation of electron–hole pair and better optical properties.

As a zero-dimensional (0D) material, graphite carbon nitride quantum dots (CNQDs) with their unique "up-conversion" properties has the ability to transform near-infrared light into visible light. Thus, CNQDs could effectively utilize both visible and infrared light in the solar spectrum and become a promising energy converter in photocatalytic systems [31,32,33,34]. Wang et al. directly prepared CNQDs using bulk C_3_N_4_ as raw material by the thermochemical etching method. The as-obtained CNQDs showed the strong blue luminescence and up-conversion performance, which could be used as an energy converter in a metal-free photocatalytic system driven by visible light [23].

According to previous literature, the construction of a g-C_3_N_4_/metal heterojunction can effectively improve the photocatalysis performance of g-C_3_N_4_ [20]. The charge transfer mechanism of a g-C_3_N_4_/metal heterojunction is different from that of g-C_3_N_4_. Due to the difference in work function, a Schottky barrier is generated at the interface between the metal and semiconductor. The presence of a Schottky barrier can significantly promote electron transfer and hence charge carrier separation. In addition, Ag and Au nanoparticles exhibit the surface plasmon resonance (SPR) effect under visible light irradiation, which can significantly promote optical absorption [35]. Compared with Au nanoparticles (Au NPs) and other precious metals, Ag NPs with excellent antibacterial activity and relatively low price have become a hot topic of g-C_3_N_4_ composite modification. The addition of Ag NPs not only expands the absorption of visible light, but also inhibits the recombination rate of photoelectron–hole pairs, thus improving the photocatalytic activity of g-C_3_N_4_. For instance, Song et al. successfully synthesized Ag/g-C_3_N_4_ composites for photocatalytic degradation of sulfamethoxazole (SMX) by the photo-reduction method. The average diameter of Ag NPs (20 nm) was well controlled when the Ag content increased from 1 to 10 wt%. Compared with pristine g-C_3_N_4_, the removal rate of SMX by 5 wt% Ag/g-C_3_N_4_ increased by 32.1%, due to the SPR effect of Ag NPs [20]. In addition, Ag NPs possess excellent antibacterial properties, which are widely applied as an antibacterial agent. Wei et al. prepared an Ag/g-C_3_N_4_ composite material for photocatalytic sterilization of Escherichia coli. The results showed that 3-Ag/g-C_3_N_4_ composites showed the best photocatalytic bactericidal effect within 120 min [36]. 

In this study, based on the above considerations, the degradation behaviors of antibiotics were systematically studied in the Ag/CNQDs/g-C_3_N_4_ photocatalytic system under visible light. The effects of initial pH and coexistence of HA and HCO_3_^-^ on the removal of different types of antibiotics (NOR, SMX and TCH) were investigated thoroughly. The degradation mechanism was investigated by the free radical quenching experiments and ESR technology. On this basis, the DFT calculation of molecular structure and LC-MS analysis were performed to explore the possible pathways of these three types of antibiotics (NOR, SMX and TCH). Furthermore, the photocatalytic degradation products of three antibiotics were detected by UV–vis and 3D EEMs. In addition, the ecological safety of antibiotic products was evaluated by the antibacterial experiment.

## 2. Results and Discussion

### 2.1. Characterization of Ag/CNQDs/g-C_3_N_4_

Figure 1a shows some characteristic diffraction peaks of the XRD pattern in the range of 10° to 80°, indicating that the Ag/CNQDs/g-C_3_N_4_ composite has a relatively stable crystal structure. In particular, two typical diffraction peaks at 2θ = 27.74° and 12.1° corresponded to the (002) and (100) peak planes of g-C_3_N_4_ (PDF#87-1526), respectively. In addition, the diffraction peaks observed at 38.18°, 44.28°, 64.46° and 77.44° matched well with the (111), (200), (220) and (311) crystal planes of standard Ag (PDF#04-0783), which indicates that the Ag/CNQDs/g-C_3_N_4_ composite was successfully prepared [37]. As demonstrated in Figure 1b, the infrared characteristic absorption peaks at 810 and 1000~1700 cm^−1^ belonged to the vibration of the s-triazine unit and C-N stretching vibration of the CN heterocyclic ring, while the wide vibration band in the range of 3000~3500 cm^−1^ was ascribed to the stretching vibration of C-OH and N-H, indicating that the Ag/CNQDs/g-C_3_N_4_ composites still retained the basic functional group structure of g-C_3_N_4_ [38]. Figure 1c shows that the as-prepared composites exhibited an ultra-thin two-dimensional origami lamellar structure. It can be seen from Figure 1d that the CNQDs were uniformly distributed on the surface of g-C_3_N_4_ nanosheets, and Ag NPs with a diameter of 5–10 nm were also successfully loaded onto the g-C_3_N_4_ nanosheets. Furthermore, the BET surface area and Barrette–Joynere–Halenda (BJH) pore diameter analyses results show that the as-obtained Ag/CNQDs/g-C_3_N_4_ composites possessed a mesoporous structure, and the pore size is mainly concentrated around 30 nm (Appendix A). Moreover, the XPS survey spectrum further confirmed the presence of C, N, O and Ag elements in the Ag/CNQDs/g-C_3_N_4_ composites without other impurities (Figure 1e). The optical properties of the synthesized Ag/CNQDs/g-C_3_N_4_ were obtained by UV–vis diffuse reflection (Figure 1f). Ag/CNQDs/g-C_3_N_4_ had a certain optical absorption in the visible light range of 400 to 800 nm, and the band gap of Ag/CNQDs/g-C_3_N_4_ was calculated to be 2.48 eV. Compared to g-C_3_N_4_ and CNQDs/g-C_3_N_4_, Ag/CNQDs/g-C_3_N_4_ could significantly promote the light absorption range due to the addition of the CNQDs with unique up-conversion characteristics and the Ag NPs with SPR effect.

### 2.2. The Ag/CNQDs/g-C_3_N_4_ Photocatalytic Degradation Efficiency of Typical Antibiotics

The photocatalytic activity of Ag/CNQDs/g-C_3_N_4_ was evaluated via the photocatalytic degradation of three types of antibiotics under visible light irradiation, and the results are shown in Figure 2. First, the concentration of antibiotics remained unchanged under the condition of photodegradation without a photocatalyst, indicating that three types of antibiotics had good photo-stability properties (Appendix A). Moreover, the Ag/CNQDs/g-C_3_N_4_ composite material could absorb up to 10% of antibiotics in the dark, indicating that the degradation of three antibiotics was mainly due to the photocatalytic reaction. As exhibited in Figure 2, compared with pure g-C_3_N_4_ and CNQDs/g-C_3_N_4_, the Ag/CNQDs/g-C_3_N_4_ composites had excellent degradation activity on the antibiotics. In the Ag/CNQDs/g-C_3_N_4_ visible light photocatalytic system, the degradation rate of NOR and SMX reached 100% and 81% within 120 min, respectively (Figure 2a,c), and the degradation rate of TCH reached 100% within 60 min (Figure 2e). In addition, as shown in Figure 2b,d,f, the photocatalytic degradation of NOR, SMX and TCH conformed to the quasi-first-order reaction kinetics, and the reaction kinetic constants (*k*) of them were 0.04233, 0.013 and 0.0735 min^−1^, respectively. The difference in the degradation rate is owed to the different structures of the three antibiotics. The dominant structure of TCH was composed of biphenyls and side groups, which were rarely found in NOR, and which were absent in SMX. Therefore, it was speculated that the oxidative active radicals produced by the Ag/CNQDs/g-C_3_N_4_ photocatalytic degradation system tended to degrade the biphenyl and side group structures of the organic pollutants. As stated above, due to the synergy of the up-conversion performance of CNQDs and the SPR effect of Ag NPs, the Ag/CNQDs/g-C_3_N_4_ composites possessed excellent photocatalytic degradation capacity of typical antibiotics.

### 2.3. The Influencing Factors of Photocatalytic Degradation for Typical Antibiotics

To investigate the degradation performance of Ag/CNQDs/g-C_3_N_4_ on NOR, SMX and TCH at the different initial pH values and the coexistence of HA and bicarbonate (HCO_3_^−^), influencing factor experiments were carried out.

#### 2.3.1. Initial pH

pH value has a significant impact on the chemical morphological structure of antibiotics in a water environment, and the effects of the initial pH on NOR degradation are shown in Figure 3a. The NOR degradation efficiencies at pH = 3, 5, 7, 9, 11 were 90%, 92%, 98%, 100% and 89%, respectively. The photocatalytic degradation performance of NOR by Ag/CNQDs/g-C_3_N_4_ was significantly improved under neutral and weak alkaline conditions (pH = 7–9), while the NOR degradation efficiency was decreased under strong acid or base conditions. These results could be explained by the electrostatic interaction between the NOR molecule and photocatalysts. According to the previous reports [39], there were three kinds of forms of NOR (p*K_a1_* = 6.10, p*K_a2_* = 8.70) under different pH conditions: NOR^+^ formed at pH < 6.10, neutral NOR^0^ or amphoteric NOR^±^ existing at 6.10 ≤ pH ≤ 8.70 and NOR^-^ existing at pH > 8.7. As shown in Appendix A, the zero point charge of Ag/CNQDs/g-C_3_N_4_ was close to 4.98. When pH < 4.98, the excess hydrogen occupied the adsorption site on the Ag/CNQDs/g-C_3_N_4_ surface. At the same time, the electrostatic repulsion between Ag/CNQDs/g-C_3_N_4_ with positively charged and NOR^+^ could lead to the reduction of surface adsorption, which weakened the photocatalytic degradation performance. When the pH > 8.70, the electrostatic repulsion between Ag/CNQDs/g-C_3_N_4_ when negatively charged and NOR^-^ increased gradually, which also led to weakening of the Ag/CNQDs/g-C_3_N_4_ photocatalytic degradation performance.

Figure 3b shows the effect of initial pH on SMX photocatalytic degradation. Ordinarily, SMX exists in three types under different pH values (p*K_a1_* = 1.86, p*K_a2_* = 5.73): positive form at pH < 1.86; neutral or amphoteric form at 1.86 ≤ pH ≤ 5.73 and negatively charged form at pH > 5.73 [40]. The photocatalytic degradation activity of SMX by Ag/CNQDs/g-C_3_N_4_ was gradually enhanced under neutral and weakly alkaline conditions (pH 7–9), which was consistent with the effect of pH on NOR degradation performance, and could also be explained by Zeta potential results of materials. 

Due to the presence of dimethylamine, phenoldiketone and tricarbonyl groups, tetracycline hydrochloride (TCH) is a typical amphoteric antibiotic [41]. Since the molecular structure of TCH is related to the pH condition, the degradation performance of Ag/CNQDs/g-C_3_N_4_ on TCH under different pH conditions was further studied. As shown in Figure 3c, the degradation efficiency of TCH was gradually enhanced under neutral and weakly alkaline conditions (pH = 7–9). When the pH value > 7.70, the electrostatic repulsion between Ag/CNQDs/g-C_3_N_4_ with negative charge and TCH^−^ or TCH^2−^ increased gradually, which also had detrimental effects on TCH photocatalytic degradation.

#### 2.3.2. Co-Existing Inorganic Ions

In general, the presence of inorganic salt ions in water can inhibit the oxidation of organic pollutants and toxic products by reacting with free radicals, thus making the water environment very complex. In our experiment, the effects of chloridion (Cl^−^) and bicarbonate (HCO_3_^−^) on NOR degradation were investigated. The results in Appendix A showed that the presence of Cl^−^ had a slight negative effect of three degrading antibiotics, which was due to the competitive adsorption between Cl^−^ and NOR (SMX, TCH) [42]. As a radical scavenger, HCO_3_^−^ could react with •OH to form •CO_3_^−^, which was more selective and less reactive than •OH. In the Ag/CNQDs/g-C_3_N_4_ photocatalytic degradation system, bicarbonate enhanced NOR, SMX and TCH degradation at concentrations from 2 to 5 mM, as shown in Appendix A. According to previous studies, the carbonate radical reacted rapidly with compounds containing readily oxidizing groups, including amino acids and aromatic anilines with electron-donating substituents. NOR is an amino acid analogue with two aromatic aniline structures. Moreover, the carbonate radical might also be a significant reactant for the oxidation of sulfur-containing compounds [43]. Another important point is that the solution pH increased after the addition of HCO_3_^−^, which could explain the improvement in photocatalytic degradation efficiency [43,44]. 

#### 2.3.3. HA

HA, as an important component of natural organic matter, plays an important role in the photocatalytic degradation of antibiotics. Typically, the presence of HA has negative influences on the photocatalytic degradation efficiency of NOR, SMX and TCH [20]. As shown in Figure 4, with the increase in HA concentration from 0 to 15 mg·L^−1^, the inhibition of HA was enhanced gradually. The inhibition of HA in the photocatalytic degradation process was mainly attributed to the following two reasons. On the one hand, HA could compete with the target pollutants for the active substance produced in the solution; on the other hand, HA could quench the photons produced during the photocatalytic process, thus eliminating the production of active species in the solution and inhibiting the degradation of antibiotics [45].

### 2.4. The Active Species in Photocatalytic Reactions

Generally, various radicals might be generated through oxidation and reduction in a reaction system. Therefore, the ESR technique was used to detect possible reaction species in the Ag/CNQDs/g-C_3_N_4_ photocatalytic reaction system. DMPO (5,5-dimethyl-1-pyrroline N-oxide) was added to a methanol solution and water solution as a radical trapping agent. As shown in Figure 5a, the DMPO-•O_2_^−^ and DMPO-•OH species were successfully detected in the medium solution after visible light irradiation of Ag/CNQDs/g-C_3_N_4_ for 2 min, while no signal appeared in the darkness. According to the signal intensity, •O_2_^−^ was the main reactive species under visible light irradiation.

To further determine the specific role of each active substance in the photocatalytic degradation process of NOR, SMX and TCH, the radical quenching experiments were also carried out, and the results are exhibited in Figure 5b–d. The presence of p-benzoquinone (p-BQ) significantly inhibited the photocatalytic degradation efficiency of antibiotics, indicating that •O_2_^−^ played a vital role in the degradation process of antibiotics, which was consistent with the ESR results. In the presence of ammonium oxalate (AO), the photocatalytic degradation efficiency decreased slightly, indicating that h^+^ was another active species. In addition, compared with p-BQ and AO, tertiary butyl alcohol (t-BuOH) had the least effect on the degradation efficiency of antibiotics, indicating that •OH was a kind of auxiliary active specie. The active species in the Ag/CNQDs/g-C_3_N_4_ system were •O_2_^−^, h^+^ and •OH. Among them, •O_2_^−^ and h^+^ affect the process of photocatalytic degradation, and the role of •OH could not be overlooked either.

### 2.5. DFT Calculation of Antibiotic Reactive Sites and Antibiotic Degradation Pathways

To clarify the degradation pathway more accurately, the natural population analysis (NPA) charge distribution and Fukui index (*f*^0^) for radicals attack of three antibiotic molecules at B3LYP/6-31 + G(d,p) were calculated by using DFT calculations. Bonds with larger *f*^0^ in antibiotic molecules were more susceptible to the radical attack. To further study the detailed degradation process of the three antibiotics, HPLC/MS was applied to detect the intermediates produced during the photocatalytic degradation process. The possible intermediates were determined from the molecular weight.

#### 2.5.1. NOR Degradation

The HPLC/MS was used to detect the molecular weight of intermediates in Ag/CNQDs/g-C_3_N_4_ photocatalytic degradation of NOR, and the deduced intermediates are shown in Table 1.

As shown in Figure 6a, the results of the theoretical calculations showed that C1, C2, C6, C8, N17, O13, O14 and F23 in the NOR structure are more vulnerable to the reactive radicals (ROs) attack. According to previous literature, the oxidized active radicals produced by a photocatalytic degradation system tend to oxidize the biphenyls and side chains of organic pollutants [46]. Based on the Fukui index combined with intermediate detection results, the degradation pathway of NOR was proposed (Figure 6b,c). In general, the initiation of NOR degradation was mainly caused by free radical attack on piperazine and quinolone groups. Figure 6c shows that defluorination, dehydrogenation, and transformation of quinolones and piperazines occurred simultaneously or sequentially. Under the free radical attack, the intermediate A (M.W. 304) was the dehydroxylation product, and the intermediate B (M.W. 318) was the defluorination product of NOR. In addition, the pheiperazine ring in NOR was another active group easily attacked by free radicals. In this step, six intermediates were mainly identified in the protonation of intermediate D (M.W. 350), all of which were formed on the piperazine ring by oxidation, ring-opening and partial elimination [47].

#### 2.5.2. SMX Degradation

Similarly, the intermediates in the Ag/CNQDs/g-C_3_N_4_ photocatalytic degradation of SMX by HPLC/MS are shown in Table 2.

Figure 7a provides the chemical structure of SMX, and Figure 7b presents the free radical attack Fukui index calculated (*f*^0^) according to the NPA charge distribution of the SMX molecule. It is shown that C1, C2, C4, C5, C6, N7, N11, O16 and O17 in SMX were more vulnerable to ROs attack. As shown in Figure 7c, the ROs including h^+^, •O_2_^−^ and •OH could decompose organic molecules by dehydrogenation, electron transfer or addition elimination. The intermediate product A (M.W. 284) was 30 Da higher than the SMX parent ion (M.W. 254), which might be an oxidizing intermediate attacked by •OH/•O_2_^−^ on the active site N7 of SMX. This is a common pathway in the degradation process of SMX and has been reported in many previous studies. Furthermore, the intermediate product B (M.W. 270) was 16 Da higher than the SMX parent compound ion, which might be the SMX hydroxylation intermediate formed in the presence of ROs. The DFT calculation indicated that the *f*^0^ of the C5 and C6 atoms in the benzene ring were 0.05 and 0.0425, respectively, which were very vulnerable to •OH/•O_2_^−^ attack. In addition, nitration of the amino group on the benzene ring was another possible degradation pathway. Finally, these small molecules might be further degraded into CO_2_ and H_2_O [20].

#### 2.5.3. TCH Degradation

Based on the most favorable sites of C2, C6, C18, O_2_7, O_2_8 and O32 in the TCH molecule (Figure 8a,b), and the intermediates detected by HPLC-MS (Table 3), the possible degradation pathway of TCH by Ag/CNQDs/g-C_3_N_4_ is shown in Figure 8c. First, the TCH was decomposed into tetracycline (TC) in aqueous solution, and the ROs easily attacked the C=C, amine and phenolic groups in the TC molecules. There are three main degradation pathways of TC during the photocatalytic oxidation process. In pathway I, the -N(CH_3_)_2_ group in the TC molecule could be gradually attacked by ROs to form intermediate C (M.W. 430). In pathway Ⅱ, the TC molecule (M.W. 445) was first hydroxylated to form hydroxylated TC (A, M.W. 461). The hydroxylated TC compound could then be further oxidized to form compound B (M.W. 424). In addition, in pathway III, compound D (M.W. 400) was first obtained by the loss of formamide in the TC molecule, and it was further attacked by ROs to form compound E (M.W. 414) [48]. The DFT calculation showed that the C17, C25, O_2_7 and O_2_8 atoms in the TCH skeleton were very fragile with *f^0^* values of 0.018, 0.0098, 0.0402 and 0.0493, respectively. Therefore, compound E (M.W. 414) was an intermediate in formamide loss and oxidation caused by the radical attack. Then, these intermediates were further oxidized to small organic compounds through functional group cleavage, intermolecular rearrangement and ring-opening reactions.

### 2.6. The Mineralization Process of Antibiotics and the Evaluation of Bacteriostatic Properties of Intermediate Products

To evaluate the mineralization process of antibiotics, the photocatalytic degradation processes of NOR, SMX and TCH were monitored by UV–vis spectrophotometry. As shown in Figure 9a, two significant peaks were observed at 275 and 325 nm in the initial NOR solution. With the 120 min photocatalytic reaction, the intensity of the two characteristic peaks gradually decreased, indicating the degradation of NOR. Seen from Figure 9b, the initial SMX had a typical characteristic absorption peak at 280 nm, and the intensity of the characteristic peak decreased gradually during the photocatalytic degradation process of 120 min, which proved the degradation of SMX. Furthermore, a new peak appeared at 320 nm, and its intensity gradually increased, indicating the formation and accumulation of intermediates. As shown in Figure 9c, the initial TCH has three characteristic absorption peaks at 250, 275 and 360 nm, respectively. In the process of photocatalytic degradation, the characteristic peaks at 250 and 275 nm gradually disappeared, and the intensity of the characteristic peak at 360 nm gradually decreased. In addition, the peak formed at 260 nm, and the intensity gradually increased, indicating the degradation of TCH and the formation of intermediates.

The degradation and mineralization of NOR, SMX and TCH during the Ag/CNQDs/g-C_3_N_4_ photocatalytic process were monitored by 3D excitation emission matrix fluorescence spectroscopy (3D EEMs), and the results are shown in Figure 10. As shown in Figure 10a, the typical fluorescence peaks at Ex/Em = 250–300/350–475 nm (I) and Ex/Em = 300–350/400–450 nm (II) of NOR were located in the humic acid region, indicating that NOR could be considered as a humic acid substance. As shown in Figure 10b, the typical fluorescence peak intensity of NOR decreased within the 120 min photocatalytic reaction, indicating that the concentration of NOR was gradually reduced. In addition, the reduction in the fluorescence signal was mainly due to the fact that the intermediates generated did not belong to humic acid substances [49]. As seen from Figure 10c,e, similar to NOR, the typical fluorescence peaks at Ex/Em = 200–300/340–440 nm in the initial SMX solution and Ex/Em = 250–350/360–480 nm in the initial TCH solution were also located in the humic acid region. Therefore, SMX and TCH could also be considered as humic acid substances. As shown in Figure 10d,f, the intensity of each peak gradually decreased, indicating that the concentration of SMX and TCH gradually reduced after photocatalytic degradation reactions. These results are consistent with the above studies.

In addition, the antibacterial activities of the intermediates produced during the photocatalytic degradation of the three antibiotics were evaluated. Using Escherichia coli as the control substance, the antibacterial activity was measured by the Oxford Cup method [50]. As a comparison, the antibacterial test results of antibiotics with only light irradiation or a photocatalyst are shown in Appendix A. It can be seen that the bacteriostatic ring still existed after only light irradiation or a photocatalyst, indicating that the antibacterial activity of antibiotics was not lost. As shown in Figure 11a, with the increase in photocatalytic time, the diameter of the bacteriostatic ring decreased from 11 to 0 mm after 120 min, indicating that the residual NOR in the water environment lost its antibacterial activity after the photocatalytic reaction. As seen from Figure 11b,c, the antibacterial zone diameters of SMX and TCH also significantly reduced, indicating that the antibacterial performance of antibiotics after photocatalytic degradation greatly decreased. These observations indicated that the NOR, SMX and TCH almost lost their antibacterial activities after the photocatalytic reaction, which was beneficial to the recycling of antibiotic wastewater.

## 3. Experimental Section

### 3.1. Synthesis of Ag/CNQDs/g-C_3_N_4_ Composite

The synthesis methods of the Ag/CNQDs/g-C_3_N_4_ composite are provided in Appendix A.

### 3.2. Characterization of Ag/CNQDs/g-C_3_N_4_ Composite

Various characterization methods including X-ray diffraction (XRD), Fourier transform infrared spectroscopy (FT-IR), scanning electron microscopy (SEM), transmission electron microscopy (TEM), X-ray photoelectron spectroscopy (XPS) and UV–vis diffuse reflectance spectroscopy (UV–vis DRS) were applied to systematically study the microstructure, crystal structure, composition and light absorption properties of Ag/CNQDs/g-C_3_N_4_ composites. In addition, the information on the use of LC-MS, ESR and 3D EEM tools and their procedures, as well as procedures for testing the antibacterial activity of intermediate products after carrying out antibiotic photodegradation tests, are listed in Appendix A.

### 3.3. Photocatalytic Tests

The specific experiment methods of photocatalysis degradation are shown in Appendix A. 

### 3.4. Theoretical Calculation

The Fukui functions were used to predict the reaction sites for electrophilic, nucleophilic, and free radical attacks [51], and the detailed information is provided in Appendix A.

## 4. Conclusions

In this study, the as-obtained Ag/CNQDs/g-C_3_N_4_ exhibited excellent photocatalytic degradation efficiency of different antibiotics (NOR, SMX, TCH), which could be ascribed to the increase in active sites in the composite photocatalyst and the synergistic effects between Ag NPs, CNQDs and g-C_3_N_4_. •O_2_^−^, h^+^ and •OH were generated in the Ag/CNQDs/g-C_3_N_4_ photocatalytic degradation system. Furthermore, the degradation effects of NOR, SMX and TCH were significantly increased under neutral and weakly alkaline conditions (pH 7–9). In addition, the presence of HA had a slight negative effect on the degradation effect. In contrast, the coexistence of appropriate HCO_3_^−^ had a positive effect on the degradation effect. The density functional theory (DFT) calculation combined with the HPLC-MS analysis predicted the possible photocatalytic degradation pathways of NOR, SMX and TCH. Finally, the antibacterial tests indicated that the toxicity of the degradation intermediates and products decreased significantly.

## Figures and Tables

**Figure 1 molecules-28-01597-f001:**
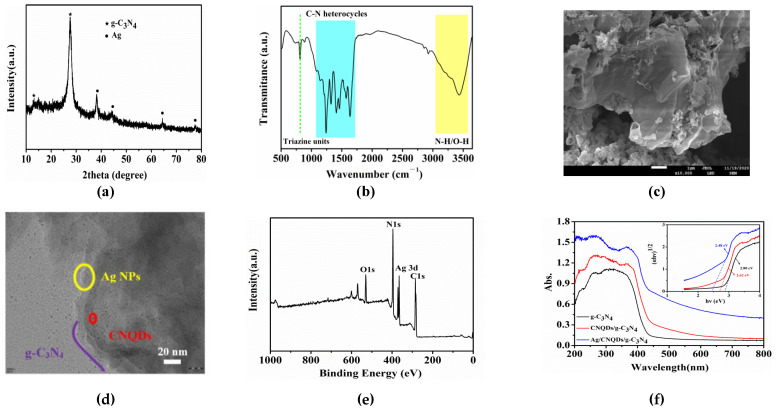
Characterization of Ag/CNQDs/g-C_3_N_4_ composites: (**a**) XRD pattern; (**b**) FTIR spectra; (**c**) SEM image; (**d**) TEM image; (**e**) XPS spectra; (**f**) UV–vis diffuse reflectance spectra.

**Figure 2 molecules-28-01597-f002:**
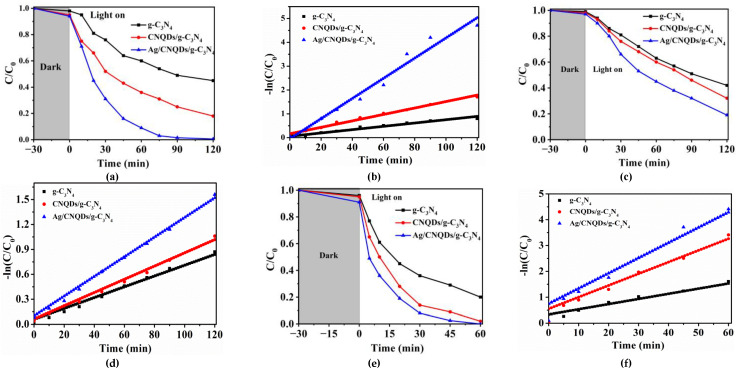
Photocatalytic degradation curves and kinetics of NOR (**a**,**b**), SMX (**c**,**d**) and TCH (**e**,**f**) by Ag/CNQDs/g-C_3_N_4_.

**Figure 3 molecules-28-01597-f003:**
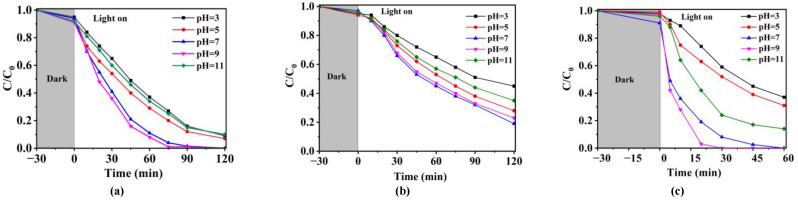
Effects of pH on the degradation efficiency of (**a**) NOR, (**b**) SMX and (**c**) TCH by Ag/CNQDs/g-C_3_N_4_ (25 °C, photocatalyst dosage = 0.2 g/L, antibiotic concentration = 10 mg/L).

**Figure 4 molecules-28-01597-f004:**
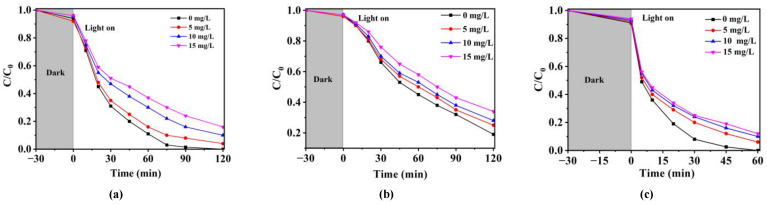
Effect of HA on the degradation efficiency of (**a**) NOR, (**b**) SMX and (**c**) TCH by Ag/CNQDs/g-C_3_N_4_ (25 °C, photocatalyst dosage = 0.2 g/L, antibiotic concentration = 10 mg/L).

**Figure 5 molecules-28-01597-f005:**
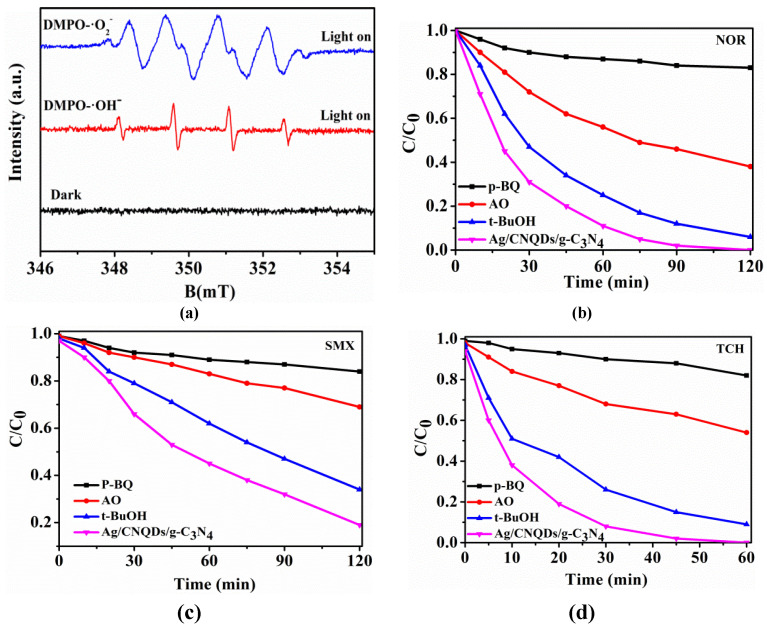
(**a**) ESR spectra of DMPO-•O_2_^–^ and DMPO-•OH. Effect of quenching agent on degradation of (**b**) NOR, (**c**) SMX and (**d**) TCH by Ag/CNQDs/g-C_3_N_4._

**Figure 6 molecules-28-01597-f006:**
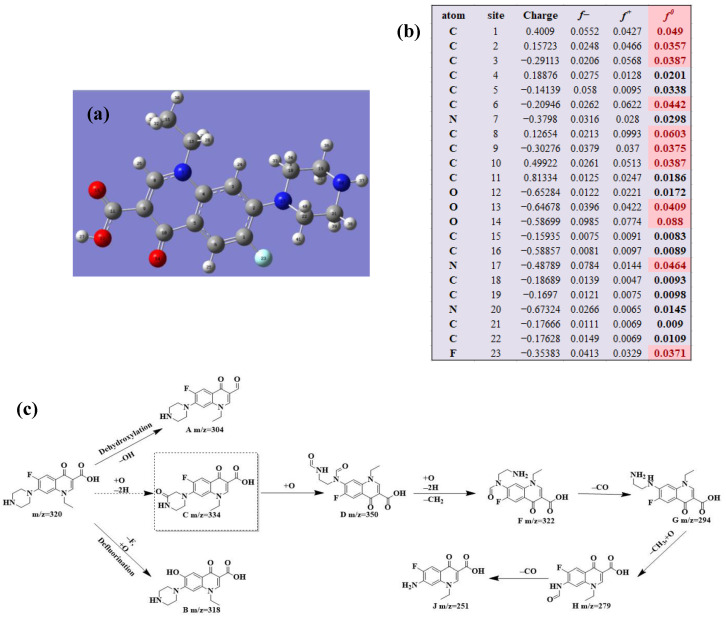
(**a**) Chemical structure, (**b**) NPA charge distribution and Fukui index (*f^0^*) and (**c**) degradation pathway of NOR.

**Figure 7 molecules-28-01597-f007:**
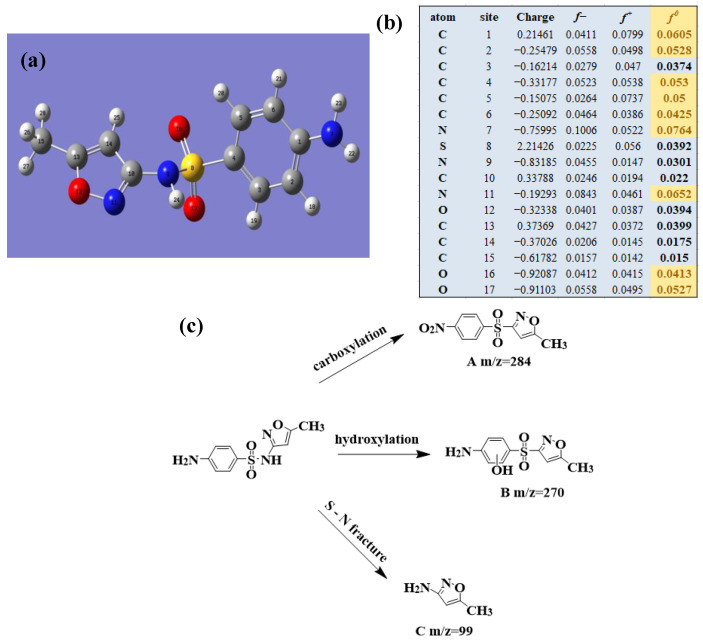
(**a**) Chemical structure, (**b**) NPA charge distribution and Fukui index (*f^0^*) and (**c**) degradation pathway of SMX.

**Figure 8 molecules-28-01597-f008:**
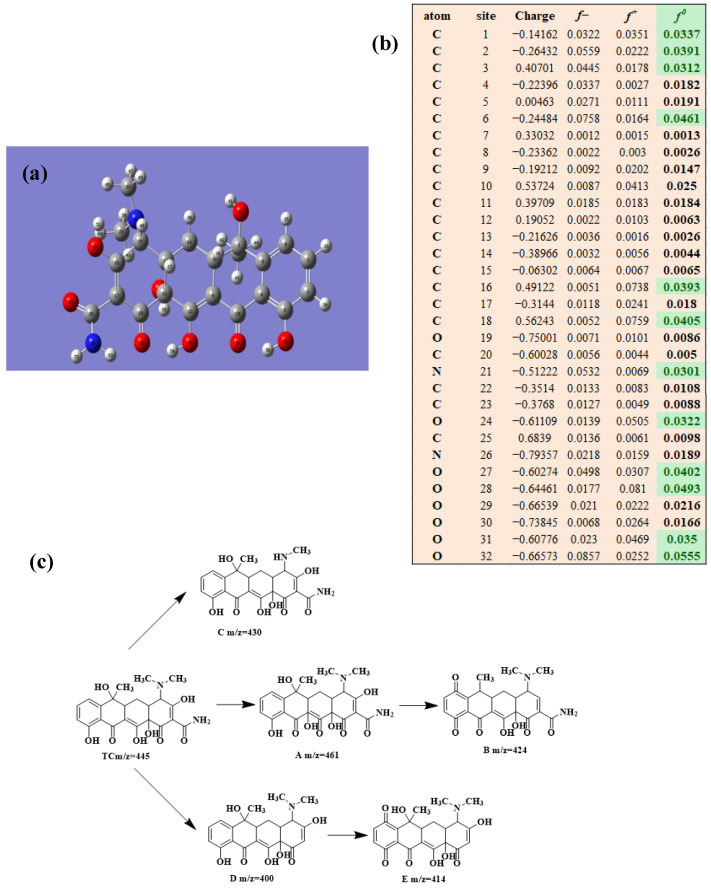
(**a**) Chemical structure, (**b**) NPA charge distribution and Fukui index (*f^0^*) and (**c**) degradation pathway of TC.

**Figure 9 molecules-28-01597-f009:**
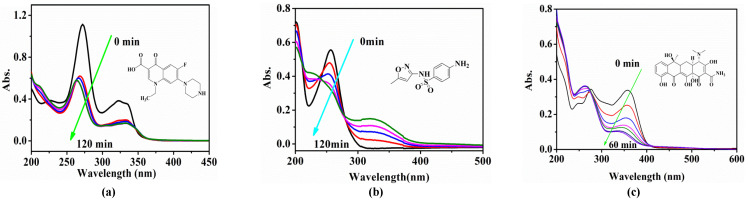
Time-dependent UV–vis spectra of (**a**) NOR, (**b**) SMX and (**c**) TCH solution for Ag/CNQDs/g-C_3_N_4_ composites.

**Figure 10 molecules-28-01597-f010:**
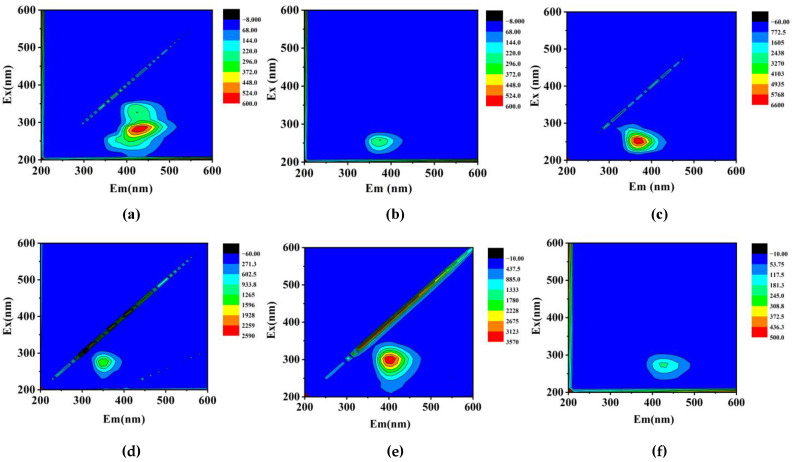
Three-dimensional EEMs images of (**a,b**) NOR and (**c**,**d**) SMX after visible light irradiation 0, 120 min; and (**e**,**f**) TCH solutions after visible light irradiation 0, 60 min.

**Figure 11 molecules-28-01597-f011:**
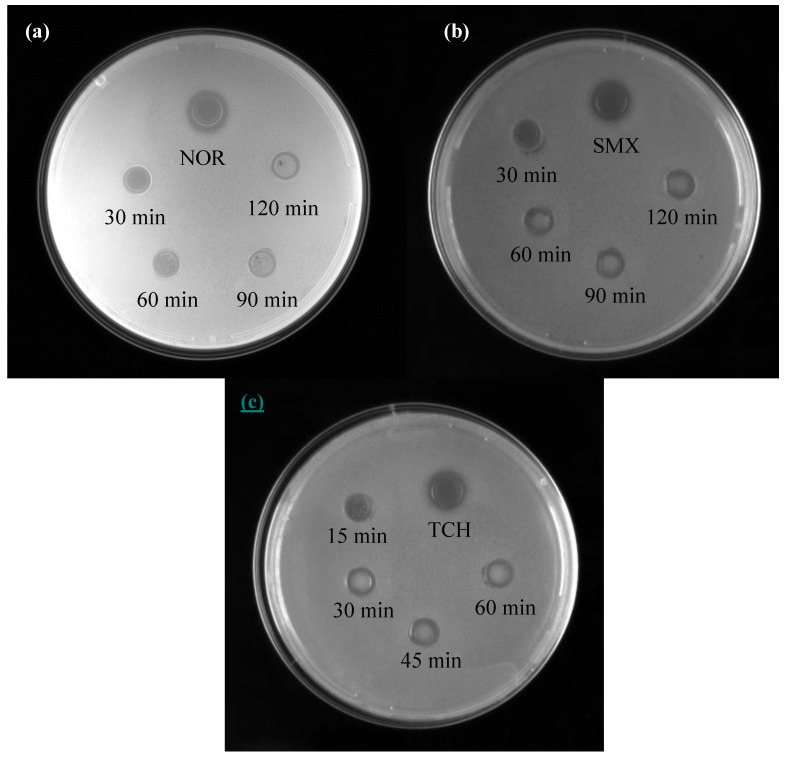
Antibacterial activities of NOR (**a**), SMX (**b**) and TCH (**c**) and their degradation products.

**Table 1 molecules-28-01597-t001:** The relative molecular weights and possible structural formulae of the NOR photo-degradation intermediates.

Compounds	Molecule	*m*/*z*	Possible Structure
NOR	C_16_H_18_FN_3_O_3_	320	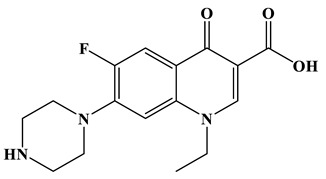
NOR-1	C_16_H_18_FN_3_O_2_	304	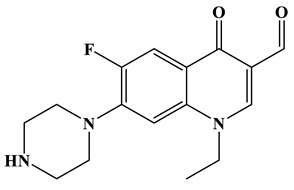
NOR-2	C_16_H_19_N_3_O_4_	318	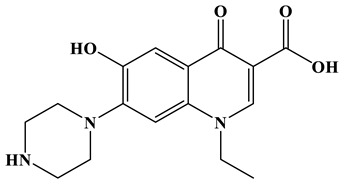
NOR-3	C_16_H_16_FN_3_O_5_	350	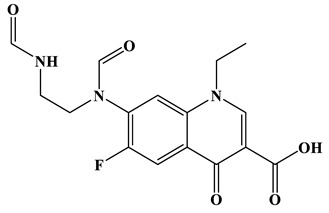
NOR-4	C_15_H_16_FN_3_O_4_	322	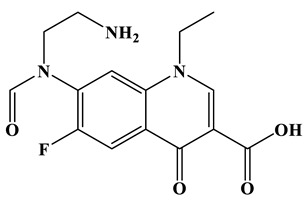
NOR-5	C_14_H_16_FN_3_O_3_	294	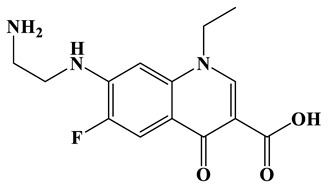
NOR-6	C_13_H_11_FN_2_O_4_	279	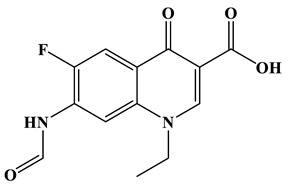
NOR-7	C_12_H_11_FN_2_O_3_	251	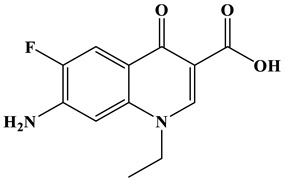

**Table 2 molecules-28-01597-t002:** The relative molecular weights and possible structural formulae of SMX photo-degradation intermediates.

Compounds	Molecules	*m*/*z*	Possible Structure
SMX	C_10_H_11_N_3_SO_3_	254	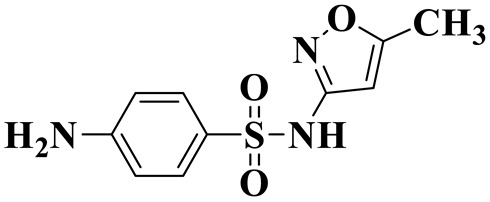
SMX-1	C_10_H_9_N_3_SO_5_	284	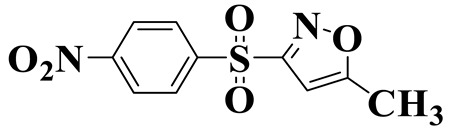
SMX-2	C_10_H_11_N_3_SO_4_	270	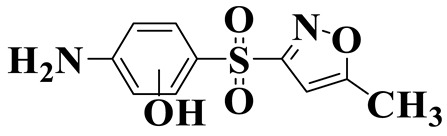
SMX-3	C_4_H_6_N_2_O	99	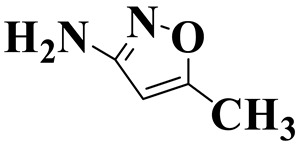

**Table 3 molecules-28-01597-t003:** The relative molecular weights and possible structural formulae of TC photo-degradation intermediates.

Compounds	Molecules	*m*/*z*	Possible Structure
TC	C_22_H_24_N_2_O_8_	445	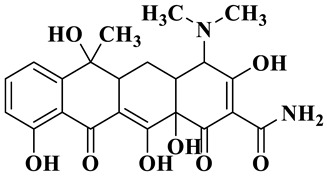
TC-1	C_22_H_24_N_2_O_9_	461	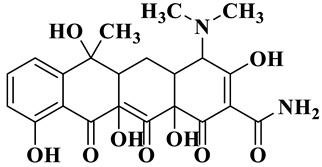
TC-2	C_21_H_22_N_2_O_8_	430	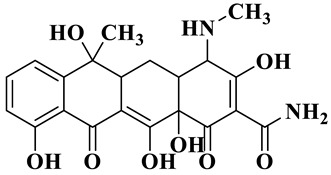
TC-3	C_22_H_38_N_2_O_7_	424	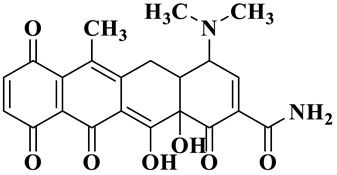
TC-4	C_21_H_23_NO_7_	400	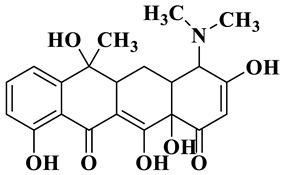
TC-5	C_21_H_21_NO_8_	414	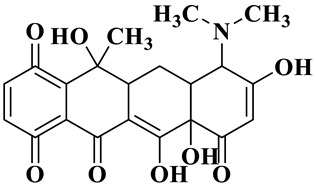

## Data Availability

The data presented in this study are available in Appendix A.

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
