# Peer review of "In-Depth Insight into the Ag/CNQDs/g-C3N4 Photocatalytic Degradation of Typical Antibiotics: Influence Factor, Mechanism and Toxicity Evaluation of Intermediates"

_molecules, 2023, doi:10.3390/molecules28041597_

Round 1
Reviewer 1 Report
The article is a complete and harmonious scientific research. There are minor stylistic remarks presented in the attached file.

Author Response
Thank you very much for your helpful comments and suggestions on our manuscript (No.: molecules-2185854). We have checked and revised our manuscript carefully according to the comments. The revised places have been highlighted in the text. Our responses to the comments are listed as follows.
- Indistinguishable scale in Figure 1 c and d.
Response: Thanks for your suggestion. The scale in Figure 1 c and d has been re-labeled (page 4).
- electron spin resonance (ESR) - transcript on page 8 and the abbreviation is already used on pages 1, 2, 4
Response: Thanks for your suggestion. The transcript of electron spin resonance (ESR) has been added on page 1 where it first appeared, and the subsequent occurrences have been changed to abbreviations.
- natural population analysis (NPA) transcript on page 12 and already in use on pages 9 and 11
Response: Thanks for your suggestion. The transcript of natural population analysis (NPA) has been added on page 9 where it first appeared, and the subsequent occurrences have been changed to abbreviations.
- reactive radicals (ROs) transcript on page 12 and already in use on page 10
Response: Thanks for your suggestion. The transcript of reactive radicals (ROs) has been added on page 10 where it first appeared, and the subsequent occurrences have been changed to abbreviations.
- reactive radicals (ROs) stands for abbreviation twice on pages 12 and 13
Response: Thanks for your suggestion. The transcript of reactive radicals (ROs) has been added on page 10 where it first appeared. ROs on pages 12 and 13 have been modified to abbreviations.
- What is DMPO? enter transcript
Response: Thanks for your suggestion. DMPO (5,5-dimethyl-1-pyrroline N-oxide) is a radical trapping agent. The transcript of DMPO has been added on page 8.
- What is HA? page 1 and 5
Enter the transcript in the text of the manuscript and not in the supplementary. This will make the article easier to read.
Response: Thanks for your suggestion. HA is the abbreviation of humic acid, and the corresponding transcript has been added on page 1.
- Correct figure 2с
Response: Thanks for your suggestion. Figure 2с has been corrected on page 5.
- Add a bond between nitrogen and oxygen or hydrogen atoms (Table 2)
Response: Thanks for your suggestion. The bond between nitrogen and oxygen atoms has been added in Table 2 (page 11).
- In Table 3 and the caption to Fig. 8c, replace TCH with TC because the Fukui index and degradation of the TC molecule and not its hydrochloride are investigated above.
Response: Thanks for your suggestion. The TCH has been replaced with TC in Table 3 and the caption to Fig. 8c (page 13, 14).

Reviewer 2 Report
The manuscript entitled “In-depth insight into the Ag/CNQDs/g-C3N4 photocatalytic degradation of typical antibiotics: Influence factor, mechanism and toxicity evaluation of intermediates” is focused on in-depth analysis of the mechanism behind the photocatalytic degradation of three widely used antibiotics (norfloxacin, tetracycline hydrochloride and sulfamethoxazole) by Ag/CNQDs/g-C3N4. Although the system per se is not something new the manuscript provides a serious analysis of the mechanism behind the catalyst activity backed by theoretical calculations. The additional antibacterial activity of the is a nice touch and shows proves the successful degradation of the antibiotics. The manuscript is well written with conclusions supported by the results. However, prior to publication in Molecules the following points should be addressed:
1. According to the Supplementary information the authors prepared 5 catalysts with different Ag loadings. Nowhere in the text (nor the supplementary information) the authors mention the effect of the Ag loading nor the way they measured the amount of Ag on the CNQDs/g-C3N4. There is just a short sentence saying that the 3% Ag was the optimal amount but there are no experimental evidence for this claim. Please add the photocatalytic experiments that include the activity of the catalysts with the different Ag loadings. Furthermore, was the activity similar towards all three antibiotics?
2. For the phase identification based on the XRD patterns please provide the number of the ICDD numbers since the JCPDS database is now outdated (and it is now part of the ICDD database).
3. Please provide a better SEM images since those are not clear and the AgNPs are hardly seen. May be detection of the backscattered electrons can provide a better contrast and will make the identification of the AgNPs clearer.
4. The authors calculated the bandgap of the prepared nanocomposite but it is unclear how they did it. Did they used the Tauc’s equation of Kubelka-Munk function? The authors should also state the type of band gap they assumed during the calculations (direct or indirect bandgap), although, it is now well known that the g-C3N4 has indirect bandgap. Furthermore please provide the bandgap of the pristine g-C3N4 and CNQDs/g-C3N4 for the sake of comparison.
5. It is unclear why the catalytic activity of the Ag/CNQDs/g-C3N4 is compared with the activity of the pristine g-C3N4 instead of CNQDs/g-C3N4. Please add the photocatalytic activity of the CNQDs/g-C3N4 as well to really evaluate the better catalytic activity of the AgNPs loaded catalysts.
Author Response
Thank you very much for your helpful comments and suggestions on our manuscript (No.: molecules-2185854). We have checked and revised our manuscript carefully according to the comments. The revised places have been highlighted in the text. Our responses to the comments are listed as follows.
- According to the Supplementary information the authors prepared 5 catalysts with different Ag loadings. Nowhere in the text (nor the supplementary information) the authors mention the effect of the Ag loading nor the way they measured the amount of Ag on the CNQDs/g-C3N4. There is just a short sentence saying that the 3% Ag was the optimal amount but there are no experimental evidence for this claim. Please add the photocatalytic experiments that include the activity of the catalysts with the different Ag loadings. Furthermore, was the activity similar towards all three antibiotics?
Response: Thanks for your suggestion. The photocatalytic experiments that include the activity of the catalysts with the different Ag loadings have been added in supplementary data (Fig. S1).
The exact Ag content of different Ag/CNQDs/g-C3N4 photocatalysts powders were determined by ICP-OES. The results of ICP-OES analysis showed that the actual content of Ag in Ag/CNQDs/g-C3N4 composites with 1 wt. %, 3 wt. %, 5 wt. %, 8 wt. % and 10 wt. % were measured to be 0.94 wt. %, 2.57 wt. %, 4.68 wt. %, 8.32 wt. % and 9.63 wt. %, respectively. As shown in Fig. S1, the 3% Ag/CNQDs/g-C3N4 possessed the best photocatalytic degradation performance.
The Ag/CNQDs/g-C3N4 composites had the excellent degradation activity on the three antibiotics. However, the degradation rates of the three antibiotics were still different. As shown in Fig. 2 (b, d, f), the photocatalytic degradation of NOR, SMX and TCH conformed to the quasi-first-order reaction kinetics, and the reaction kinetic constants (k) of them were 0.04233 min-1, 0.013 min-1 and 0.0735 min-1, respectively. The difference of the degradation rate was mainly due to the difference structure of the three antibiotics. The dominant structure of TCH, consisting of biphenyls and side groups, was rarely found in NOR, but absent in SMX. Therefore, it was speculated that the oxidative active radicals produced by the Ag/CNQDs/g-C3N4 photocatalytic degradation system tended to degrade the biphenyls and side group structures of organic pollutants.
- For the phase identification based on the XRD patterns please provide the number of the ICDD numbers since the JCPDS database is now outdated (and it is now part of the ICDD database).
Response: Thanks for your suggestion. The number of the stand numbers of g-C3N4 and Ag NPs based on the XRD patterns have been updated on page 3 through literature review.
- Please provide a better SEM images since those are not clear and the Ag NPs are hardly seen. May be detection of the backscattered electrons can provide a better contrast and will make the identification of the Ag NPs clearer.
Response: Thanks for your suggestion. A better SEM images has been added in Fig. 1c (page 4).
- The authors calculated the bandgap of the prepared nanocomposite but it is unclear how they did it. Did they used the Tauc’s equation of Kubelka-Munk function? The authors should also state the type of band gap they assumed during the calculations (direct or indirect bandgap), although, it is now well known that the g-C3N4 has indirect bandgap. Furthermore please provide the bandgap of the pristine g-C3N4 and CNQDs/g-C3N4 for the sake of comparison.
Response: Thanks for your suggestion. The Kubelka-Munk function was used to calculate the bandgap of the prepared nanocomposites. It is assumed that all the as-prepared nanocomposites have indirect band gap during the calculations. Furthermore, the bandgap of the pristine g-C3N4 and CNQDs/g-C3N4 have been provided in Fig. 1f, and the corresponding explanation has been added on page 3.
Compared to g-C3N4 and CNQDs/g-C3N4, the Ag/CNQDs/g-C3N4 could significantly promote the light absorption range due to the addition of the CNQDs with the unique up-conversion characteristics and the Ag NPs with SPR effect.
- It is unclear why the catalytic activity of the Ag/CNQDs/g-C3N4 is compared with the activity of the pristine g-C3N4 instead of CNQDs/g-C3N4. Please add the photocatalytic activity of the CNQDs/g-C3N4 as well to really evaluate the better catalytic activity of the Ag NPs loaded catalysts.
Response: Thanks for your suggestion. The photocatalytic activities of the CNQDs/g-C3N4 against three typical antibiotics have been added in Fig. 2 (page 5) for better comparison.

Reviewer 3 Report
I recommend this article to be accepted with minor revision

Author Response
Dear Editor and Reviewers,
Thank you very much for your helpful comments and suggestions on our manuscript (No.: molecules-2185854). We have checked and revised our manuscript carefully according to the comments. The revised places have been highlighted in the text. Our responses to the comments are listed as follows.
Response to Reviewer:
Introduction:
Suggestion: quantitative data on the degradation performance of antibiotics using other catalysts used as references
Response: Thanks for your suggestion. The degradation performance of antibiotics using other catalysts have been added in the Introduction part for references.
Typo:
In page 2: As a zero dimensional (0 D) materials, graphite carbon nitride quantum dots () with
Response: Thanks for your suggestion. The error on page 2 has been corrected.
In page 4: Fig. 1e, Binging Energy …
Response: Thanks for your suggestion. The error in Fig. 1e has been corrected (page 4).
In Experimental section:
In page 3 on Photocatalytic tests, particularly in Information text S2 : have not mentioned the quantity of catalyst used !
Response: Thanks for your suggestion. The specific process of photocatalysis experiment has been provided in Supporting information Text S3 (the original Text S2).
Typically, the photocatalyst dosage was 0.2 g/L.
It is suggested to add information on the use of LC-MS, ESR and 3D EEM tools and their procedures, as well as procedures for testing the antibacterial activity of intermediate products after carrying out antibiotic photodegradation tests.
Response: Thanks for your suggestion. The information on the use of LC-MS, ESR and 3D EEM tools and their procedures, as well as procedures for testing the antibacterial activity of intermediate products after carrying out antibiotic photo-degradation tests have been listed in supplementary data (text S2).
In Results and Discussion
It is suggested to add the result of the antibacterial test of antibiotics without irradiation and catalyst treatment, to find out the difference.
Response: Thanks for your suggestion. The antibacterial test of antibiotics with only light irradiation or photocatalyst have been added in supplementary data (Fig. S7).
As a comparison, the antibacterial test results of antibiotics with only light irradiation or photocatalyst were shown in Fig. S7. It can be seen that the bacteriostatic ring still existed after only light irradiation or photocatalyst, indicating that the antibacterial activity of antibiotics was not lost. As shown in Fig.11 (a), with the increase of photocatalytic time, the diameter of bacteriostatic ring decreased from 11 mm to 0 mm after 120 min, indicating that the residual NOR in water environment lost its antibacterial activity after the photocatalytic reaction. Seen from Fig. 11 (b, c), the antibacterial zone diameters of SMX and TCH also significantly reduced, indicating that the antibacterial performance of antibiotics after photocatalytic degradation greatly decreased. These observations indicated that the NOR, SMX and TCH almost lost their antibacterial activities after the photocatalytic reaction, which was beneficial to the recycling of antibiotic wastewater.
Regarding the particle size of the Ag/CNQDs/g-C3N4 composites described in Fig. 1 c-d should be given statistical size distribution data.
Response: Thanks for your suggestion. The BET surface area and Barrette-Joynere-Halenda (BJH) pore diameter analyses results have been added in supplementary data (Fig. S2), and the corresponding explanation has been added on page 3.
As shown in Fig. S2, the as-obtained Ag/CNQDs/g-C3N4 composites had the mesoporous structure, and the pore size is mainly concentrated around 30 nm.

Round 2
Reviewer 2 Report
Dear authors thank you for addressing all of my questions and comments. I believe the manuscript is now suitable for publishing in Molecules.